# Platelet Count/Spleen Diameter Ratio as a Non-Invasive Predictor of Esophageal Varices in Cirrhotic Patients: A Single-Center Experience

**Srinith Patil** [1], **Swarup Kumar Patnaik** [2], **Manjit Kanungo** [2], **Kanishka Uthansingh** [2], **Jimmy Narayan** [2], **Subhasis Pradhan** [2], **Debakanta Mishra** [2], **Manoj Kumar Sahu** [3] and **Girish Kumar Pati** [2,*]

1   Department of Gastroenterology, ESIC Medical College & Hospital, Kalaburagi 585106, India
2   Department of Gastroenterology, Institute of Medical Sciences and SUM Hospital, Siksha O Anusandhan University, Bhubaneswar 751003, India; jimmynarayan@soa.ac.in (J.N.); debakanta87@gmail.com (D.M.)
3   Department of Gastroenterology, Apollo Hospital, Bhubaneswar 751005, India; manojsahu427@gmail.com
*   Correspondence: girishkumarpati@soa.ac.in

**Abstract:** (1) Background: The current study examined the correlations between platelet count (PC), spleen diameter (SD), and their ratio to establish a non-invasive technique for predicting the presence of oesophageal varices in cirrhotic patients. (2) Methods: The current study was an observational study conducted in the Gastroenterology Department at IMS and SUM Hospital from November 2019 to November 2021. Consecutive cirrhotic patients without a history of gastrointestinal bleeding were enrolled in the study, and the esophageal varices were assessed. The patients underwent the necessary tests, including upper gastrointestinal endoscopy, liver function testing, abdominal ultrasonography, and full hemograms. All these parameters were analyzed statistically through SPSS version 23, and $p \leq 0.05$ was considered statistically significant. (3) Results: There were significant differences between cases with and without esophageal varices in the following parameters: PC, SD and their ratio, hemoglobin, and ALT level. The PC/SD ratio of $\leq 1400$ was associated with a sensitivity of 90.9%, specificity of 80.8%, and a positive predictive value of 82.56% in predicting the presence of oesophageal varices, as per receiver operating curve (ROC) analysis in our study. (4) Conclusions: Esophageal varices can be predicted non-invasively using the platelet count, spleen diameter, and PC/SD ratio.

**Keywords:** esophageal varix; portal hypertension; platelet count; spleen diameter

## 1. Introduction

Cirrhosis cases can present with many clinical manifestations [1]. Its characteristic features include substantial hepatic fibrosis, irreversible hepatic parenchymal destruction, and formation of regenerative nodules. Gilbert first coined the term "portal hypertension" to characterize the occurrence of increased spontaneous blood flow, leading to vasodilation and increased hepatic resistance due to cirrhotic liver [2]. The esophageal variceal development is among the significant consequences of portal hypertension (PHT) [3]. Approximately one-third of patients succumb to gastroesophageal variceal bleed [4]. As per Baveno VII consensus on PHT, patients with cirrhosis should be subjected to an upper gastrointestinal endoscopic (UGI) test to assess the presence of oesophageal varices at diagnosis [5]. Repeat UGI endoscopy is recommended every 1–2 years and 2–3 years for patients with small varices and no varix, respectively, to assess the disease progression [6].

One of the significant limitations of the endoscopy study is that it is an invasive and uncomfortable diagnostic procedure with limited availability and might not be affordable and feasible in some cases due to associated comorbidities and financial constraints [7], which demand availability of non-invasive, easily available and affordable tools, with significant predictive value in esophageal variceal diagnosis. Non-invasive tools such

as platelet count/spleen diameter (PC/SD) ratio, fibrotest, and fibroscan test were used in prior studies for esophageal variceal prediction; however, PC/SD ratio was found to be most promising [8]. The rationale for carrying out the current study is 'better clinical outcomes can be observed by early detection of esophageal varices with timely effective management in needy patients'. The esophageal varices are classified as Grade I: straight and unbendable; Grade II: tortuous, occupying < 1/3rd of the esophageal lumen; and Grade III: large, occupying > 1/3rd of the esophageal lumen [9]. Previous reports suggested various non-invasive diagnostic markers for the early prediction of oesophageal varices [9–20]. The non-invasive markers for esophageal variceal prediction described were PC, prothrombin time (PT), albumin concentration, splenic size, and portal vein diameter (on ultrasound) [14]. Previous reports suggested that thrombocytopenia, splenomegaly, and ascites can all independently predict the presence of large oesophageal varices in cirrhotic patients with a higher risk for bleeding [21,22]. Esophageal varices initially appear only when the hepatic venous pressure gradient (HVPG) is >10 Hg. Its size, ranging from small to large, increases by 5 to 10% per year, and its increasing size, with associated increased variceal-wall tension, leads to variceal rupture and bleeding [14]. Analyzing the above study results, it can be hypothesized that, by using the above noninvasive parameters, UGI endoscopy might be deferred in cases with a lesser probability of high-grade esophageal varices, and endoscopy-related patient discomfort, the financial and endoscopic workload can be shortened remarkably [22]. The platelet count < 88,000/cc mm can effectively predict the presence of large oesophageal varices with an odds ratio (OR) of 5.5 and 95% CI (1.8–20.6) and gastric varices with OR of 5 and 95%CI (1.4–23) [23]. Previous reports recommended that following the first UGI endoscopy study, in cases with low-grade esophageal varices, PC/SD ratio assessment can be used as an effective screening tool to track the esophageal variceal progression [24].

Observing the above scenario, the current study was conducted with the aim to find out the relationship between platelet count, spleen diameter, and their ratio with the presence of oesophageal varices in cirrhotic patients without prior history of gastrointestinal (GI) bleeding.

## 2. Materials and Methods

The present study is a descriptive observational single-center cross-sectional study carried out in the Department of Gastroenterology, IMS and SUM Hospital, Bhubaneswar, India.

### 2.1. Data Collection Methods

Consecutive patients with cirrhosis fulfilling the inclusion criteria from November 2019 to November 2021 were included in the study and evaluated with appropriate, feasible tests. This study comprised cirrhotic cases without any prior history of upper or lower gastrointestinal bleeding. Cirrhosis diagnosis can be objectively confirmed by ultrasonography (USG) abdomen study, which usually reveals the presence of irregular liver margin, coarse architecture, dilated portal veins and collaterals, shrunken liver size, enlarged spleen, dilated splenic vein and collaterals [22]. Also, a hemogram reveals the presence of thrombocytopenia, LFT shows the presence of reversal of albumin (A)to globulin(G) ratio, low albumin level, aspartate transaminase (AST) to alanine transaminase (ALT) ratio > 1, and raised international normalized ratio (INR) prothrombin time (PT), and UGI Scopydemostrates presence of dilated esophageal veins or varices in cases with cirrhosis [22].

Cases with underlying conditions were excluded from the study such as: history of usage of beta blocker, procoagulants or anticoagulant drugs; esophageal varices with history of sclerotherapy or band ligation; transjugular intrahepatic portosystemic shunt (TIPS) or portal hypertension surgery. This study was approved by the institutional ethics committee, and valid written consent was obtained from each participant prior to inclusion in the study.

The patients were subjected to routine investigations such as complete hemograms, including hemoglobin, total white cell count, differential count, and total platelet count (TPC). They were also subjected to detailed liver function tests (LFT), including serum bilirubin, albumin, ALT, AST, alkaline phosphatase, and INR-PT tests. USG abdomen and pelvis studies were performed to assess the liver architecture, portal hypertension, and measurement of bipolar spleen diameter objectively. A UGI endoscopy study was carried out to evaluate and grade the esophageal varices. The spleen enlargement was subjectively assessed manually (Tables 1 and 2) and objectively evaluated by ultrasonography study (Table 3) for better characterization and visibility. The PC/SD ratio was calculated for every patient in the study. The study group was divided into two groups, including cases with and without esophageal varices, and different parameters were compared between them.

**Table 1.** Baseline demographic data of all the patients.

| Baseline Parameters | Study Population (N = 125) | |
|:---:|:---:|:---:|
| | **Mean $\pm$ SD** | **Median (IQR)** |
| Age (years) | 53.85 $\pm$ 12.52 | 53 (45–63) |
| Hb (g/dL) | 10.99 $\pm$ 2.42 | 11 (9–13) |
| PC(/cu mm) | 122,935 $\pm$ 59,402 | 112,000 (94,500–138,000) |
| SD (mm) | 116.83 $\pm$ 25.54 | 121 (93.5–137) |
| PC/SD | 1195.85 $\pm$ 864.10 | 963 (749–1283) |
| TB (mg/dL) | 6.26 $\pm$ 6.28 | 4.30 (2.6–8.6) |
| DB (mg/dL) | 4.14 $\pm$ 4.89 | 2.6 (1.65–4.85) |
| AST (IU/L) | 75.2 $\pm$ 45.47 | 65 (42.5–98) |
| ALT (IU/L) | 72.1 $\pm$ 55.84 | 61 (31.5–101) |
| TP (gm/dL) | 6.71 $\pm$ 0.83 | 6.8 (6.2–7.4) |
| Alb (gm/dL) | 2.71 $\pm$ 0.6 | 2.7 (2.4–3.2) |
| ALP (IU/L) | 205 $\pm$ 126 | 177 (98.5–287) |
| PT (s) | 23.13 $\pm$ 8.34 | 21 (17.45–27.26) |
| INR | 1.43 $\pm$ 0.60 | 1.27 (1–1.80) |

SD: standard deviation; IQR: interquartile range; Hb: hemoglobin; PC/SD: platelet count (PC), spleen diameter (SD), subjectively assessed; TB: total bilirubin; DB: direct bilirubin; AST: aspartate aminotransferase; TP: total protein; Alb: albumin levels; ALP: alkaline phosphatase; PT: prothrombin time (PT); INR: international normalized ratio.

**Table 2.** Biochemical parameters and PC/SD distribution among the two groups.

| | Varices | N | Mean $\pm$ SD | Median (IQR) | *p* Value * |
|:---:|:---:|:---:|:---:|:---:|:---:|
| Hb (g/dL) | Present | 99 | 10.713 $\pm$ 2.3 | 10.6 (9–13) | 0.023 |
| | Absent | 26 | 12.019 $\pm$ 2.6 | 12.5 (9.7–14) | |
| PC | Present | 99 | 107,434.3 $\pm$ 45,308.5 | 104,000 (90,000–123,000) | <0.001 |
| | Absent | 26 | 181,957.7 $\pm$ 69,857.8 | 183,000 (146,500–206,250) | |
| SD | Present | 99 | 121.62 $\pm$ 25 | 125 (100–138) | <0.001 |
| | Absent | 26 | 98.62 $\pm$ 19.1 | 93 (88.5–101.5) | |
| PC/SD | Present | 99 | 985.3162 $\pm$ 730.44 | 873.13 (725.92–873.13) | <0.001 |
| | Absent | 26 | 1997.467 $\pm$ 876.46 | 1994.62 (1502.5–2455.6) | |
| TB | Present | 99 | 5.973 $\pm$ 6.15 | 3.8 (2.4–7.8) | 0.132 |
| | Absent | 26 | 7.396 $\pm$ 6.76 | 6.05 (2.88–9.25) | |
| Age | Present | 99 | 54.93 $\pm$ 12.64 | 55 (48–64) | 0.033 |
| | Absent | 26 | 49.73 $\pm$ 11.34 | 47.5 (43–55) | |
| DB | Present | 99 | 4.0141 $\pm$ 4.76 | 2.6 (1.5–5.1) | 0.391 |
| | Absent | 26 | 4.6392 $\pm$ 5.43 | 2.85 (2.06–4.76) | |
| AST | Present | 99 | 72.02 $\pm$ 39.89 | 65 (43–96) | 0.321 |
| | Absent | 26 | 87.46 $\pm$ 61.82 | 82.5 (42–103.26) | |
| ALT | Present | 99 | 64.55 $\pm$ 40.54 | 58 (25–97) | 0.016 |
| | Absent | 26 | 100.73 $\pm$ 89.1 | 96 (51.75–112.5) | |
| TP | Present | 99 | 6.794 $\pm$ 0.8 | 6.9 (6.2–7.4) | 0.026 [#] |
| | Absent | 26 | 6.385 $\pm$ 0.9 | 6.5 (5.73–7.1) | |

**Table 2.** *Cont.*

|  | Varices | N | Mean ± SD | Median (IQR) | *p* Value * |
|---|---|---|---|---|---|
| Alb | Present | 99 | 2.682 ± 0.5 | 2.6 (2.3–3.1) | 0.078 |
|  | Absent | 26 | 2.835 ± 0.6 | 3.1 (2.6–3.3) |  |
| ALP | Present | 99 | 205.81 ± 126 | 177 (101–287) | 0.631 |
|  | Absent | 26 | 200 ± 130 | 136.5 (88.25–299.9) |  |
| PT | Present | 99 | 23.796 ± 7.7 | 22 (19–27.6) | 0.015 |
|  | Absent | 26 | 20.577 ± 10 | 19.8 (13.76–22.5) |  |
| INR | Present | 99 | 1.4817 ± 0.6 | 1.31 (1.1–1.8) | 0.104 |
|  | Absent | 26 | 1.2458 ± 0.6 | 1.05 (0.8–1.27) |  |

* Mann–Whitney U *p*-value; # independent samples '*t*' test *p*-value. SD: standard deviation; IQR: interquartile range; Hb: hemoglobin; PC/SD: platelet count (PC), splenic diameter (SD): subjectively assessed; TB: total bilirubin; DB: direct bilirubin; AST: aspartate aminotransferase; TP: total protein; Alb: albumin levels; ALP: alkaline phosphatase; PT: prothrombin time (PT); INR: international normalized ratio.

**Table 3.** Correlation of grade of varices with SD, PC, and PC/SD.

| Parameters |  | Grade of Varices |  |  |  |  |  |  |  | Total | *p*-Value |
|---|---|---|---|---|---|---|---|---|---|---|---|
|  |  | Nil |  | I |  | II |  | III |  |  |  |
|  |  | n | % | n | % | n | % | n | % |  |  |
| SD | 100–150 | 19 | 15.2 | 13 | 10.4 | 4 | 3.2 | 6 | 4.8 | 42 |  |
|  | 150–200 | 7 | 5.6 | 22 | 17.6 | 34 | 27.2 | 20 | 16 | 83 |  |
|  | Total | 26 | 20.8 | 35 | 28 | 38 | 30.4 | 26 | 20.8 | 125 | 0.735 |
| PC | 50,000–100,000 | 3 | 2.4 | 9 | 7.2 | 15 | 12 | 20 | 16 | 47 |  |
|  | 100,000–150,000 | 3 | 2.4 | 22 | 17.6 | 19 | 15.2 | 6 | 4.8 | 50 |  |
|  | >150,000 | 20 | 16 | 4 | 3.2 | 4 | 3.2 | 0 | 0 | 28 |  |
|  | Total | 26 | 20.8 | 35 | 28 | 38 | 30.4 | 26 | 20.8 | 125 | <0.001 |
| PC/SD ratio | 500–1000 | 2 | 1.6 | 14 | 11.2 | 23 | 18.4 | 17 | 13.6 | 56 |  |
|  | 1000–2000 | 11 | 8.8 | 16 | 12.8 | 10 | 8 | 5 | 4 | 42 |  |
|  | >2000 | 13 | 10.4 | 5 | 4 | 5 | 4 | 4 | 3.2 | 27 |  |
|  | Total |  | 26 |  | 35 |  | 38 |  | 26 | 125 | <0.001 |

SD: spleen bipolar diameter, measured objectively ultrasonographically; PC: platelet count.

## 2.2. Statistical Analysis

A descriptive statistical analysis was performed to analyze the data generated in the current study. The results of categorical variables are shown in number (%). However, the results of continuous variables are displayed as mean ± SD (Min–Max) and median (Interquartile range). For non-skewed data, the independent Student's *t*-test was employed; for skewed data, the Mann–Whitney U test was utilized. The receiver operating characteristic (ROC) curve was generated to access PC, PC/SD ratio, SD cut-off values, and their respective sensitivity and specificity. One-way ANOVA is used in multivariate regression analysis to determine the significant differences. '*p*' value less than ≤0.05 is defined as significant. SPSS version 23 was employed for the statistical analysis.

## 3. Results

A total of 125 cirrhotic patients were included in the study, and their mean age of presentation was 53.85 ± 12.52 years, their average PC was 122,935 ± 59,402/cu mm, and their mean SD (subjectively assessed) was 116.83 ± 25.54 mm as narrated in the baseline demographic data in Table 1.

In the current study, out of a total 125 cases, 79.2% were males and 20.8% were females; 86% cases had ascites and 79.2% cases had varices. The comparison of various biochemical markers and PC/SD were assessed subjectively between the two groups; cases with and without varices are presented in Table 2.

Hemoglobin (Hb) level, PC, and SD were significantly different among the two groups. Lower PCs and higher mean SD values were observed in cases with varices. The PC/SD

ratio was significantly lower in cases with varices than those without varices. Patients with and without varices had a mean age of presentation of 54.9 years and 49.3 years, respectively. Significantly lower ALT levels and higher total protein (TP) levels were observed in patients with varices compared to cases without varices. Interestingly, no significant differences in the albumin values, ALP values, PT values, and INR values were noticed between the two groups.

The study population was grouped based on the PC/SD ratio, and it was found that the majority of the patients had a ratio of 500–1000 (44.8%) followed by a ratio of 1001–2000 in 33.6% of subjects, as shown in Figure 1.

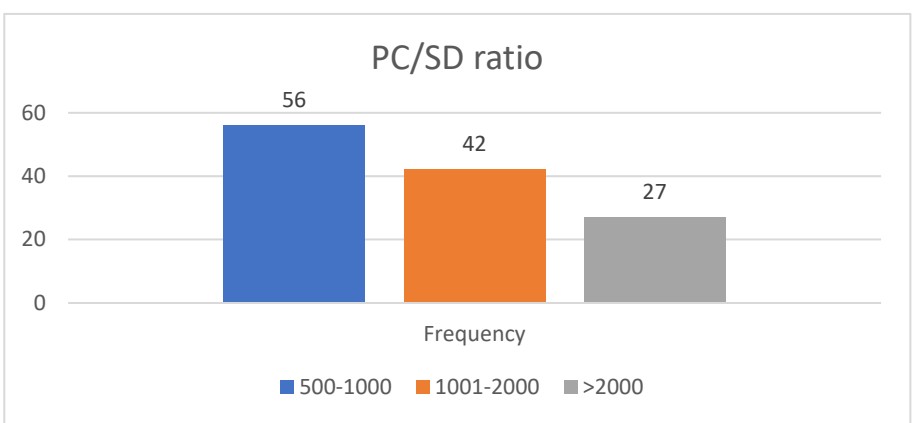

**Figure 1.** Distribution of PC/SD ratio; PC: platelet count; SD: spleen bipolar diameter.

We observed that the majority of the patients had an SD of 150–200 mm, measured objectively by ultrasonography study, and many patients (40%) had a PC of 100,000–150,000. The severity of varix grades is compared in relation to SD, PC and PC/SD in Table 3.

The ROC for PC/SD and PC and ROC for SD are depicted in Figures 2 and 3, respectively.

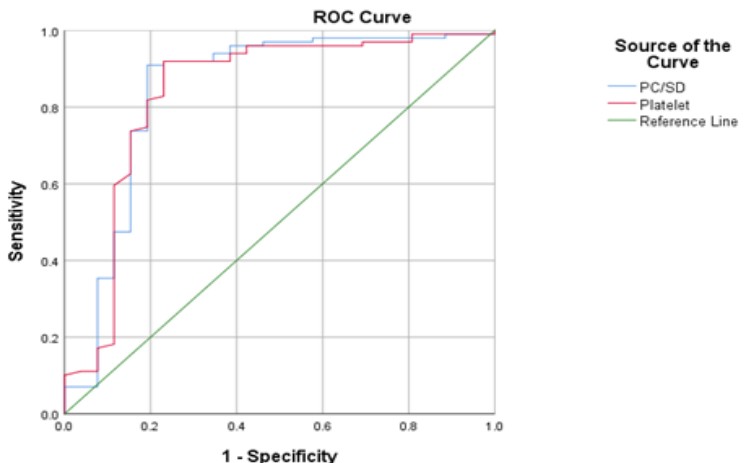

**Figure 2.** ROC for PC/SD and PC. Blue: PC/SD; Red line: TPC; Green line: Reference line.

To distinguish between patients with and without varices, the cut-off values for the PC/SD ratio, PC, and SD were determined using the ROC curve analysis. The area under the curve (AUC) for PC/SD, PC, and SD was found to be 0.84, 0.837, and 0.769, respectively, which was significant with a *p*-value = 0.001. The cut-off values for PC/SD, PC, and SD by ROC curve analysis were 1400, 138,500/cu mm, and 95 mm, respectively, which can differentiate cases with varices from cases without varices. The sensitivity of PC/SD, PC, and SD cut-off values was 90.9%, 89.9%, and 81.8%, respectively, and specificity was 80.8%, 77%, and 69.2%, respectively. Varices are, therefore, more likely to occur in patients with PC/SD, a PC below the cut-off value, and an SD above the cut-off value. The

positive predictive value of the cut-off values was calculated to be 82.56%, 79.56%, and 72.64%, respectively.

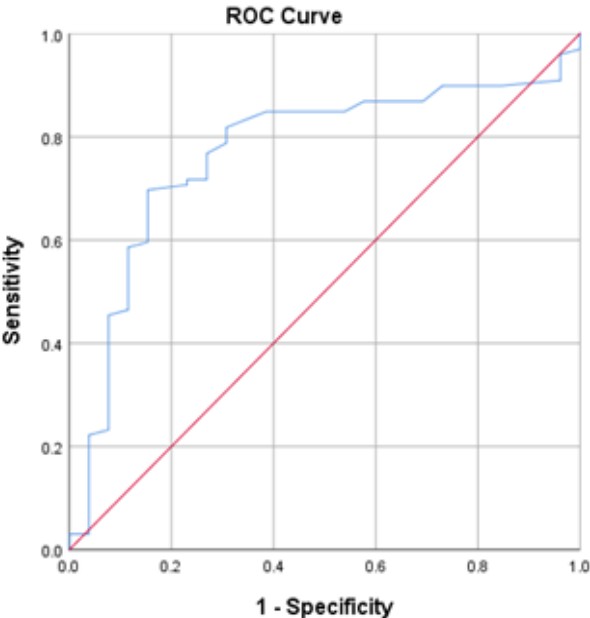

**Figure 3.** ROC for SD. Red line: Referemc e line; Blue line: SD.

## 4. Discussion

Massive upper gastrointestinal bleeding due to portal hypertension (PHT)-related variceal bleeding can affect approximately 30–40% of cirrhotic populations, leading to a number of morbidities, mortalities, and medical expenses [25]. Although a UGI endoscopy study is the best modality to diagnose and stratify esophageal varices as per Baveno VII consensus recommendation on portal hypertension [5], few noninvasive tests can predict it remarkably well. PHT can reduce the mean arterial pressure due to hyperdynamic circulation and may activate the neurohormonal mechanisms by increasing adequate circulating blood volume, out of which sympathetic nervous system (SNS) activation is proportional to the degree of splanchnic vasodilation and the functional stage of cirrhosis [26]. The study by Miceli et al. on 'Heart rate variability association with disease severity and portal hypertension in cirrhosis' is well-conducted, supporting the above statement [26]. The author suggested that autonomic nervous system (ANS) imbalance and adrenergic tone alteration in cirrhotic cases represent a perpetuated adaptation process with the ongoing progress of portal hypertension with a resultant increase in variceal development, severity, and risk of bleeding [26]. The authors concluded that the rate variability (which was used as an indirect and noninvasive measure of the degree of ANS activity proportional to the severity of PHT) may be a significant noninvasive predictor for higher variceal bleeding risk [26]. Simple, reproducible measures, such as SD, PC, and the PC/SD ratio for esophageal varices prediction, were employed in the current study. Thrombocytopenia has been shown in several studies to be an essential risk factor for variceal development and progression [10,17,21]. The cases without varices had a mean PC of 128,500/cu mm, which was more remarkable than those with small varices (107,800/cu mm) [15]. Also, a PC < 90,000/cu mm raises the risk of esophageal varix (EV) by about 2.5 times [23]. Prior reports suggested that the presence of large EV was independently associated with PC < 88,000/cu mm [15,27]. As SD is inversely correlated with PC and has been involved in the pathogenesis of thrombocytopenia in cirrhotic patients, the PC/SD ratio was postulated to be among the suitable predictors for variceal assessment. This ratio could be used to improve and stabilize the predictive value since the platelet count by itself may not always be indicative of PHT and can be deceptive, too. In 145 cirrhotic patients, it was reported that the negative predictive value of a PC/SD ratio of 909 was 100% in the

prediction of esophageal varices [13]. A PC/SD ratio cut-off value of 1014 gave a positive and negative predictive value of 95.4% and 95.1%, respectively [24]. In our study, most (40%) of the patients had a PC between 100,000 and 150,000/mm$^3$, whereas 37.6% had a PC < 100,000/mm$^3$ with a median PC of 112,000/mm$^3$ (94,500–138,000/mm$^3$). Most (66.4%) of the patients had SDs between 150 and 200 mm, and 79.2% had esophageal varices. Most (44.8%) cases had PC/SD ratios between 500 and 1000 with a median PC/SD ratio of 963 (749–1283) in our study.

By univariate analysis, significant differences ($p < 0.05$) were observed when comparing age, Hb, PC, SD, PC/SD ratio, ALT, TP, and PT between cases with and without esophageal varix. However, when comparing the grade of esophageal varices with SD, PC and PC/SD ratio, only PC and PC/SD ratio could significantly predict the variceal grades and severity by multivariate logistic regression analysis. In patients with esophageal varices, 37.6% of cases had PCs between 100,000 and 150,000/cu mm, 60.8% of cases had SDs between 150 and 200 mm, and 40.11% of cases had PC/SD ratios between 500 and 1000. The current study resulted in a sensitivity of 52.5% and a specificity of 84.6% when taking into account the PC/SD ratio cut-off value of 909, as considered by Gianni et al. [13], which produced 100% sensitivity and 93% specificity in their research. The comparative analysis of studies with a PC/SD ratio of 909, taken as a standard reference, is described in Table 4.

**Table 4.** Comparison of studies with PC/SD ratio of 909 as standard reference.

| Study | PC/SD Ratio | Sensitivity (%) | Specificity (%) |
|---|---|---|---|
| Gianni et al. [13] | 909 | 100 | 93 |
| Baig et al. [24] | 909 | 80 | 89 |
| Sarangapani et al. [28] | 909 | 88.5 | 83 |
| Schwarzenberger et al. [27] | 909 | 80 | 40 |
| Present study | 909 | 52.5 | 54.6 |

PC: platelet count; SD: spleen diameter.

A systematic review and meta-analysis by Chawla et al. yielded a pooled sensitivity of 89% and a pooled specificity of 74% when considering the PC/SD ratio of 909 as the standard reference for variceal prediction [29]. The data quality now available was not robust enough to support the PC/SD ratio of 909 as a conventional cut-off reference value for prediction, even if pooled results produced acceptable test results. It may be helpful to include other clinical features in a prediction model, such as different PC/SD cut-off values and other predictive modalities. However, the PC/SD ratio assessment seems less complicated and more affordable than other non-invasive predictive methods.

The sensitivity and specificity of our investigation, which used a PC/SD ratio cut-off value of 1400, were 90.9% and 80.8%, respectively, and the positive predictive value was 82.56%. Furthermore, the median PC/SD ratio was 873; a ratio less than 873 denoted the existence of more excellent grades of esophageal varices.

## 5. Limitations

The limitations of this study are small sample sizes, lack of internal and external validation, and single-center evaluation, from which the findings cannot be extrapolated geographically all over the world.

## 6. Conclusions

With the use of PC and the PC/SD ratio, it is possible to predict the presence and grade of esophageal varices non-invasively. The sensitivity of the PC/SD cut-off value of ≤1400 was 90.9%, and the positive predictive value was 82.56%. However, a lack of worldwide validation makes it unreliable and too early to take the esophagogastroduodenoscopy (EGD) position as the significant screening technique for predicting oesophageal varices. From the current study, it can be concluded that lower PC and lower PC/SD ratio can be associated with a higher grade of esophageal varix ($p < 0.05$); however, higher SD

can predict the presence of esophageal varix ($p < 0.05$), but cannot expect its grade and severity($p > 0.05$).

**Author Contributions:** Conceptualization, G.K.P. and M.K.S.; methodology, S.P. (Srinith Patil); software, K.U.; validation, S.K.P., M.K. and J.N.; formal analysis, K.U.; investigation, G.K.P.; resources, M.K.S.; data curation, M.K.; writing—original draft preparation, S.P. (Srinith Patil); writing—review and editing, S.P. (Subhasis Pradhan); visualization, D.M.; supervision, G.K.P.; project administration, Girish Kumar Sahu; funding acquisition, M.K.S. All authors have read and agreed to the published version of the manuscript.

**Funding:** This research received no external funding.

**Institutional Review Board Statement:** The study was approved by the Institutional Ethics Committee (IEC), IMS and SUM Hospital, Bhubaneswar. The IEC, IMS and SUM Hospital reviewed and discussed the following documents and submitted the documents on 19 June 2020. The documents reviewed were protocol synopsis dated 19 June 2020, ICF in English, and ICF in Odiya. The approval IEC letter no: Ref.no/DRI/IMS.SH/SOA/2021/030 dated 27 February 2021.

**Informed Consent Statement:** Informed consent was obtained from all subjects involved in the study.

**Data Availability Statement:** Data generated during the study may be obtained from the corresponding author with valid reason.

**Acknowledgments:** We thank the Dean and Medical superintendent who availed us the opportunity to conduct the current study. We thank Nihar Ranjan Panda who supported us for statistical analysis during the revision process of the study. Moreover, we thank the endoscopy and OPD staff who supported us during the study.

**Conflicts of Interest:** The authors declare no conflicts of interest.

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
