# Peer review of "Platelet Count/Spleen Diameter Ratio as a Non-Invasive Predictor of Esophageal Varices in Cirrhotic Patients: A Single-Center Experience"

_gastroent, doi:10.3390/gastroent15010007_

Round 1

Reviewer 1 Report

Comments and Suggestions for Authors

The authors have analyzed platelet count, spleen diameter and their ratio as predictors for varices in cirrhotic patients. The following points may please be addressed.

Upper GI endoscopy is an office procedure and is readily available these days even in smaller cities as an essential screening tool for esophageal varices in cirrhotic patients. Hence the authors’ statement based on a 2003 reference (7) that ‘endoscopy procedure is an invasive, cumbersome, uncomfortable, and costly diagnostic procedure with limited availability, and may not be affordable and feasible to some patients because of other associated comorbidities and financial constraints’ may not be entirely accurate in present times

In the statement ‘despite its low sensitivity of 52.5% in predicting the presence of esophageal varices’ in the abstract, do the authors mean specificity? The previous sentence talks about a 90.9% sensitivity.

If this is a prospective study, the authors can mention so in the methods

Since age, ALT, total protein and PT (but not INR) were also significantly different in patients with and without esophageal varices, did the authors consider including these parameters in their analysis in addition to platelet count and spleen diameter?

As per table 2, the maximum spleen diameter is 138 (? mm) while in table 3 the range is from 100-150 and 150-200 (? 20 cm). Why this discrepancy in data?

Even so, as per table 3, around 30% of patients with spleen diameter between 10-15 cm have varices and more than 20% of patients with a platelet count of >150,000 and a platelet count/spleen diameter ratio of > 2000 have varices. Is there any way of identifying this subset of patients in clinical practice? If there is no universal screening policy for varices during first presentation in cirrhotic patients, will not these patients be missed and present with a life-threatening bleed?

The authors could discuss if platelet count and spleen diameter predict a bleed due to esophageal varices rather than just the presence of varices.

Author Response

Comment 1: Upper GI endoscopy is an office procedure and is readily available these days even in smaller cities as an essential screening tool for esophageal varices in cirrhotic patients. Hence the authors’ statement based on a 2003 reference (7) that ‘endoscopy procedure is an invasive, cumbersome, uncomfortable, and costly diagnostic procedure with limited availability, and may not be affordable and feasible to some patients because of other associated co-morbidities and financial constraints’ may not be entirely accurate in present times

Response 1: Thank you for pointing this out. We Agree with you respected sir; however, herein in India, a developing non-affluent country that is in peripheral primary health care centers in rural regions with higher population density, compared to affluent urban regions with relatively lesser population density, the availability of endoscopy and trained endoscopist is sparse due to resource constraint factor and also due to lack of awareness and phobia among the rural community about the endoscopy procedure and its associated peri-procedural choking sensation, less no. of patients opt for endoscopy despite of desperate need for this; however our statement might not be entirely accurate for other regions globally in current time due to geographic diversity, affluence and broader awareness in a different community, which is lacking in our resource constraint region with limited awareness and availability of facilities and trained per capita, physician.

Comment 2: In the statement ‘despite its low sensitivity of 52.5% in predicting the presence of esophageal varices in the abstract, do the authors mean specificity? The previous sentence talks about a 90.9% sensitivity.

Response 2: Dear sir, you have rightly pointed out our unintended mistake, in fact we mean to say specificity is 80.8% and it is not low sensitivity of 52.5%; which was wrongly mentioned in the text and we modified the text in revised version with highlighting the necessary changes for your kind scrutinization.

Comment 3: If this is a prospective study, the authors can mention so in the methods

Response 3: Sir, as this study is a cross-sectional observational study, and is not a prospective study; we have already mentioned the type of study under the material section in the text previously.

Comment 4: Since age, ALT, total protein, and PT (but not INR) were also significantly different in patients with and without esophageal varices did the authors consider including these parameters in their analysis in addition to platelet count and spleen diameter?

Response 4: Sir, you are right as age, ALT, total protein, and PT (but not INR) were also significantly different in patients with and without esophageal varices; however, we have not considered including these parameters in our analysis in addition to platelet count and spleen diameter purposefully, as these are not our targeted objectives in the current study.

Comment 5: As per table 2, the maximum spleen diameter is 138 (? mm) while in Table 3 the range is from 100-150 and 150-200 (? 20 cm). Why this discrepancy in data?

Response 5: Dear sir, in Table 2, we have narrated the palpable subjective spleen diameter below the lower coastal border, whereas in Table 3 for an accurate representation of spleen diameter objectively, we have considered the ultrasonographically measured bipolar diameter, which is presumed to be more accurate compared to palpable spleen diameter, that’s why there is discrepancy in narration of spleen diameter. We have narrated and highlighted this under the legend section beneath Table 2 and Table 3 and in the revised version of the manuscript for your reference.

Comment 6: Even so, as per Table 3, around 30% of patients with spleen diameter between 10-15 cm have varices, and more than 20% of patients with a platelet count of >150,000 and a platelet count/spleen diameter ratio of > 2000 have varices. Is there any way of identifying this subset of patients in clinical practice? If there is no universal screening policy for varices during the first presentation in cirrhotic patients, will not these patients be missed and present with a life-threatening bleed?

Response 6: Dear sir, you have rightly raised the concern; truly speaking measurement of spleen diameter and platelet count and their ratio can’t outweigh the importance of endoscopy, and can’t substitute the role of endoscopy as the primary modality of prediction of esophageal varices, and by adopting this, some cases may be missed and bleed too; however, the measurement of hepatic venous portal gradient (HVPG), wherever facility available and endoscopy might be efficient enough in prediction of esophageal varices and risk of bleeding from them.

Comment 7: The authors could discuss if platelet count and spleen diameter predict a bleed due to esophageal varices rather than just the presence of varices.

Response 7: Sorry dear Sir, as our objective is not focused on finding out the predictor of bleeding risk from esophageal varices in the current study, we are unable to discuss it in the current study; however, we are thankful to you for kind appreciation of our unbiased non manipulated original work.

Reviewer 2 Report

Comments and Suggestions for Authors

In this study, Patil et al aimed to describe the relationship between platelet count, spleen diameter, and their ratio (PC/SD) with the presence of oesophageal varices in cirrhotic patients without a prior history of upper gastrointestinal bleeding. The study, even if it lacks originality, is well conducted and described. There are below just a few issues, that were found relevant.

-       The introduction is a little bit long. I suggest reducing it and focusing on essential information about portal hypertension and the importance of noninvasive tools for esophageal varices (EV) bleeding prediction. Moreover, some information is also repeated in the first part of the discussion.

-       you affirmed, “A combination of the patient’s history, clinical manifestations, abnormal coagulation profile, impaired liver function tests, and abdominal ultrasound examination were used to diagnose the cirrhosis”. It is too generic. Please insert the specific criteria you used to define the diagnosis of cirrhosis with opportune reference.

-       Portal hypertension, and consequently EVs is the result of a hemodynamic readaptation mediated by the sympathetic nervous system. Unfortunately, no hemodynamic information about the blood pressure and heart rate of patients is presented. If you can provide this information please include it in the general parameters of the patient's presentation. At least, I suggest citing the importance of sympathetic nervous system adaptation in these patients and the possible role of some noninvasive autonomic markers together with the ones you presented [Miceli G, Calvaruso V, Casuccio A, Pennisi G, Licata M, Pintus C, Basso MG, Velardo M, Daidone M, Amodio E, Petta S, Simone F, Cabibbo G, Di Raimondo D, Craxì A, Pinto A, Tuttolomondo A. Heart rate variability is associated with disease severity and portal hypertension in cirrhosis. Hepatol Commun. 2023 Feb 9;7(3):e0050. doi: 10.1097/HC9.0000000000000050]

-       In Table 3 it is not clear if some of the Correlations of grade of varices with spleen diameter, platelet count, and PC/SD are statistically significant or not. Please provide this information in the text and add p values in Table 3.

-       In the conclusions, you cannot affirm that higher spleen bipolar diameter can be associated in your study with the presence and higher grade of esophageal varices. You just demonstrated a statistically significant association with the presence of varices but no p-values were shown about the higher grade of varices.

Author Response

Comment 1: - The introduction is a little bit long. I suggest reducing it and focusing on essential information about portal hypertension and the importance of noninvasive tools for esophageal varices (EV) bleeding prediction. Moreover, some information is also repeated in the first part of the discussion.

Response 1: Ok respected sir, as per your suggestion; we will concise the introductory part, however, the current study objective is not to access the esophageal bleeding prediction risk; rather only to access the possibility of esophageal varices along with its grade. Therefore, we have not included the data describing the importance of non-invasive tool in esophageal variceal bleeding prediction. Moreover, as rightly suggested by you, we will delete the repetitions in the initial part of the discussion, which has already been described in the introductory part of the revised manuscript.

Comment 2: - You affirmed, “A combination of the patient’s history, clinical manifestations, abnormal coagulation profile, impaired liver function tests, and abdominal ultrasound examination was used to diagnose the cirrhosis”. It is too generic. Please insert the specific criteria you used to define the diagnosis of cirrhosis with opportune reference.

Response 2: Sir, as per your suggestion, we will insert specific criteria to define the diagnosis of cirrhosis with opportune reference as required in the revised article.

Comment 3: - Portal hypertension, and consequently EVs is the result of a hemodynamic readaptation mediated by the sympathetic nervous system. Unfortunately, no hemodynamic information about the blood pressure and heart rate of patients is presented. If you can provide this information please include it in the general parameters of the patient's presentation. At least, I suggest citing the importance of sympathetic nervous system adaptation in these patients and the possible role of some noninvasive autonomic markers together with the ones you presented [Miceli G, Calvaruso V, Casuccio A, Pennisi G, Licata M, Pintus C, Basso MG, Velardo M, Daidone M, Amodio E, Petta S, Simone F, Cabibbo G, Di Raimondo D, Craxì A, Pinto A, Tuttolomondo A. Heart rate variability is associated with disease severity and portal hypertension in cirrhosis. Hepatol Commun. 2023 Feb 9;7(3):e0050. doi: 10.1097/HC9.0000000000000050]

Response 3: Sir, though we have measured vital parameters including blood pressure, heart rate, respiratory rate, the temperature in each case, however, we have not kept these data in the Excel sheet, as their assessment is not falling under the current objectives; therefore we are sorry, in current context we can’t reproduce or, manipulate the data. Further, we can’t provide the same in the revised version of the article; however, as per your suggestion, we will cite the importance of sympathetic nervous system adaptation in these patients and the possible role of some noninvasive autonomic markers together with the ones you suggested [Miceli G, Calvaruso V, Casuccio A, Pennisi G, Licata M, Pintus C, Basso MG, Velardo M, Daidone M, Amodio E, Petta S, Simone F, Cabibbo G, Di Raimondo D, Craxì A, Pinto A, Tuttolomondo A. Heart rate variability is associated with disease severity and portal hypertension in cirrhosis. Hepatol Commun. 2023 Feb 9;7(3):e0050. doi: 10.1097/HC9.0000000000000050] in the revised article.

Comment 4: - In Table 3 it is not clear if some of the Correlations of grade of varices with spleen diameter, platelet count, and PC/SD are statistically significant or not. Please provide this information in the text and add p values in Table 3.

Response 4: Sir, as rightly suggested by you, In table 3 it is not clear if some of the Correlations of the grade of varices with spleen diameter, platelet count, and PC/SD are statistically significant or not, therefore we have calculated the p-values by multivariate analysis and provided this information in the text along with p-values in Table 3 in the revised article.

Comment 5: - In the conclusions, you cannot affirm that higher spleen bipolar diameter can be associated in your study with the presence and higher grade of esophageal varices. You just demonstrated a statistically significant association with the presence of varices but no p-values were shown about the higher grade of varices.

Response 5: Dear sir, as rightly suggested by you, we will describe the p-value for different grades of esophageal varices and their clinical significance in the revised version of the article in the conclusion and discussion section. Sir, we are thankful to you for your kind appreciation of our unbiased non manipulated original work.

Reviewer 3 Report

Comments and Suggestions for Authors

Dear authors:

The topic is interesting but your works lacks novelty as there was more research articles published evaluating the same score. It would be interesting to increase the number of patients, have external validation and, if possible, compare it with the fibroscan performance, as it is the non-invasive tool more often used.

However, congratulations for your good work.

Comments on the Quality of English Language

The language needs a minor polishing please.

Author Response

Comment 1: The topic is interesting but your works lacks novelty as there was more research articles published evaluating the same score. It would be interesting to increase the number of patients, have external validation and, if possible, compare it with the fibroscan performance, as it is the non-invasive tool more often used.

Response 1: Respected sir, we agreed with you, lot more researches were available depicting the same findings like our study, however, we thought of this study in this region, because in India, a developing Nation with lesser affluence compared to the contemporary Western region and with existence of huge population density, lesser availability of higher end procedures such as endoscopy services and in particular scarcity of proportionate trained physicians  and endoscopists per needy population; compelled us for formulation of this study in our resource constrained region as this is the need of the hour for us, which can at least predict that, the patient needs endoscopy desperately at higher centre to avoid untoward consequences  but this scenario does not hold good for other global regions with better availability of above mentioned factors; however in future as per your suggestion we will try to increase the number of patients, have external validation by multicentre involvement and, will compare it with the fibroscan performance, as  a non-invasive tool, which is not possible now as we have already completed the study and can’t manipulate the data, but in future certainly we will look forward for it, therefore kindly excuse us currently as we can’t provide the same as per your requirement, however we are thankful to you for kind appreciation of our unbiased non manipulated original work.

Round 2

Reviewer 1 Report

Comments and Suggestions for Authors

Thank you for addressing all comments. 

The practical utility of using platelet count, spleen diameter (on ultrasound) and their ratio as a screening tool is to decide on the need for upper GI endoscopy to evaluate for varices. 

In resource poor situations where endoscopy may not be available, the negative predictive value of these parameters becomes important. There is still a percentage (around 20%) of patients whose varices may be missed if upper GI endoscopy is not done based on cut off values provided in this paper.

I request you to highlight these in the discussion so that it puts the utility of these simple tests in perspective for the readers.

Congratulations on a good analysis of the data

Author Response

Author’s Reply to the Review Report (Reviewer 1)

Reviewers comment: Please provide a point-by-point response to the reviewer’s comments and either enter it in the box below or upload it as a Word/PDF file. Please write down "Please see the attachment." in the box if you only upload an attachment

Author’s response:

Thank you very much for taking the time to review our manuscript. Thank you so much for each reviewer's excellent and insightful scientific comments. We have worked hard to bring scientific excellence to our manuscript. The comments raised by the reviewers were genuine and scientific; after the author's reply, the audience could understand the revised manuscript better.  The point raised initially by the first reviewer regarding the interventional procedure of endoscopy, cost constraints,s, and its availability was explained. Furthermore, the sensitivity of predicting the presence of esophageal varices and a few clarifications were mentioned after the suggestive comments. Moreover, the measurement of spleen diameter and platelet count and their ratio can’t outweigh the importance of endoscopy and can’t substitute the role of endoscopy as primary modality of prediction of esophageal varices, and by adopting this, some cases may be missed and bleed too; however, the measurement of hepatic venous portal gradient (HVPG), wherever facility available and endoscopy might be efficient enough in prediction of esophageal varices and risk of bleeding from them. All of those comments by three different reviewers were more relevant and crucial factors for our manuscript which we attempted to reply scientifically to all the reviewers.

Reviewer's comment: Please check that all references cited in the text are relevant to the manuscript's contents.
Author’s response: We reviewed the manuscript and its references. Each reference is according to the manuscript requirement.

Reviewer comment: Please provide a short cover letter detailing your changes to the
editors’ and referees’ approval.

Author’s response: Yes provided

Reviewer comment: If one of the referees has suggested that your manuscript undergo extensive English revisions, please address this issue during revision.

Author’s response: The article has been edited by our subject specialist and by an English expert for language clarification.